# Posterior Fossa Stereotactic Biopsy with Leksell Vantage Frame—Case Series and Review of Literature

**DOI:** 10.3390/jcm14020609

**Published:** 2025-01-18

**Authors:** Hojka Rowbottom, Rok Končnik, Janez Ravnik, Tomaž Šmigoc

**Affiliations:** Department of Neurosurgery, University Medical Centre Maribor, 2000 Maribor, Slovenia; hojka.rowbottom@ukc-mb.si (H.R.); janez.ravnik@ukc-mb.si (J.R.)

**Keywords:** stereotactic frame, brain biopsy, posterior fossa, preoperative planning, neuronavigation, neurosurgery

## Abstract

**Background:** Stereotactic biopsy of posterior fossa lesions, which are often inoperable, enables a safe trajectory and provides tissue samples for accurate diagnosis, which is crucial for correct treatment since the latest World Health Organization Classification of Tumors of the Central Nervous System from 2021 places immense emphasis on molecular diagnostics. Stereotactic biopsy using the Leksell Vantage headframe is, due to its rigid design, extremely accurate, but stiffer, making the procedure more challenging and the learning curve steeper. **Methods:** This retrospective analysis demonstrates the introduction of the new Leksell Vantage headframe in day-to-day practice at the University Medical Center in Maribor, Slovenia, in demanding procedures of posterior fossa biopsies, and also provides a review of the literature available on the topic with emphasis on the technical aspect of posterior fossa biopsy using the Leksell Vantage headframe in adults. **Results:** In the observed series of three patients with posterior fossa lesions, all biopsies were representative, despite tissue samples being small, providing conclusive histopathologic reports (glioblastoma, rosette-forming glioneuronal tumor and metastasis of melanoma) with additional molecular diagnostics. After the initial biopsy case, the preoperative planning times and procedure times were shortened as we learnt about the importance of a tailored approach from the first case. In all cases, the biopsy was performed under local anesthesia with patients being awake throughout surgery. **Conclusions:** The rigid Leksell Vantage headframe makes access to the posterior fossa tougher when compared to its predecessors. However, the procedure is very accurate but requires precise preoperative planning and a customized approach when placing the headframe.

## 1. Introduction

### 1.1. Background 

The posterior fossa, extending from the tentorium cerebelli superiorly to the foramen magnum inferiorly, housing the cerebellum, pons and medulla, is where approximately 10 to 15% of all brain tumors in the adult population are located [1]. A biopsy of the posterior fossa and brainstem lesions, which to this day remains challenging, is indicated when surgical resection is not feasible [2,3,4]. Stereotactic biopsy of posterior fossa neoplasms enables a trajectory that samples all relevant tumor sites as well as the tumor-infiltrating zone, contrast-enhancing locations and suspected necrotic areas, providing histological material in the quickest and safest manner [5,6]. Lesions located below the cerebral peduncles should be accessed infratentorially and lesions situated in the peduncles should be accessed supratentorially [7,8]. The suboccipital approach allows for a short trajectory, which minimizes the risks of damage to healthy brain tissue [9]. Open biopsy for deep-seated tumors is outdated [10,11]. 

Despite advances in imaging methods, magnetic resonance imaging (MRI) alone cannot reliably distinguish between the three most common diagnoses of posterior fossa lesions (brain tumor, lymphoma, inflammatory process) [12]. The MRI-based glioma grading and classification has an accuracy of less than 35%, hence, obtaining a tissue sample is vital, since the 5th edition of the World Health Organization (WHO) Classification of Tumors of the Central Nervous System (CNS) from 2021 places great emphasis on molecular diagnostics [5,13,14]. 

Biopsy of CNS neoplasms can be performed by craniotomy, burr hole, twist drill or endoscopic techniques, typically using neuronavigation or stereotaxy [15]. Frame-based stereotactic biopsy of the posterior fossa is associated with restricted entry and deep target points; however, it provides high-accuracy tissue samples with few complications [5,8,12,16]. Commercially available frames, namely Leksell G, Riechert Mundinger, Brown-Roberts-Wells and Cosman-Roberts-Wells allow for adjustments to make posterior fossa biopsy easier [9,17,18,19,20]. The Leksell Vantage frame (Elekta, Stockholm, Sweden), introduced in 2016, has a rigid design, which can only be positioned in the anteroposterior fashion, thus increasing stability, but compromising flexibility, making posterior fossa biopsy more demanding [16,21]. Its predecessor, the Leksell G-frame, could be positioned upside down to expand the access window to make posterior fossa biopsy less challenging [16]. 

### 1.2. Objectives 

In this article, we want to demonstrate our experience and the learning curve with posterior fossa biopsies using the Leksell Vantage frame as well as compare it to other frames and discuss the relevant literature focusing on posterior fossa and brainstem biopsies. 

## 2. Methodology 

Patients who underwent posterior fossa biopsy using the Leksell Vantage headframe at the Department of Neurosurgery at the University Medical Centre in Maribor, Slovenia were identified from a computerized operative database (Medis). Demographic and clinical information was obtained from operative and clinical case notes, pathology reports and radiological images. All patients who underwent stereotactic biopsy of the posterior fossa were included. We collected data on the duration of the preoperative planning and positioning of the Leksell Vantage headframe, which was performed at the department, the duration of surgery and the number of corrections required during surgery to obtain viable tissue samples. 

Additionally, we searched the PubMed database, using the keywords “posterior fossa” and “biopsy” and “Leksell Vantage”; only two articles were returned, one by Krüger et al. [16] and the other by Prilop et al. [22]. Both articles were included in our review, further demonstrating the rarity of our work. We then expanded our PubMed search by only using the keywords “posterior fossa” and “stereotactic biopsy”, which provided us with 122 articles; however, when changing the range to the last 20 years (January 2004 to October 2024) we found 81 articles and subsequently reduced those to 20, since the others included either incorrect patient population (children), the biopsy technique was not comparable to ours (robot-assisted biopsy), it proved impossible to retrieve full papers, or the article was not focusing on the technical aspect of posterior fossa biopsy with the Leksell Vantage headframe.

## 3. The Case Series

### 3.1. Case 1

A 69-year-old man presented with acute weakness in his lower limbs that had led to several falls. An MRI showed a lesion in the brainstem with radiological features of glioma and a contrast-enhanced MRI, performed 7 days later, showed no lesion dynamics. Six months later, he was examined in the neurosurgical outpatient clinic and the control MRI showed a slight enlargement of the lesion in the brainstem with no hydrocephalus; however, the patient was ambulatory with no new neurologic deficit, therefore, the decision was made to monitor further since the lesion was inoperable. After 3 months, the control MRI demonstrated an enlarged ventricular system and progress of the tumor in the posterior part of the brainstem and cerebellum; the lesion grew from 2.4 cm in diameter to 4.7 cm. The patient had increasing difficulties with cognitive function, walking and balance. As the tumor was inoperable, a biopsy was indicated. We performed a stereotactic biopsy of the lesion in the posterior fossa. The control CT, performed on the first postoperative day, showed a small hematoma in the biopsied location as well as a persistent hydrocephalus; thus, a ventriculoperitoneal shunt (VPS) was inserted, which led to an improvement in the patient’s higher cognitive functions and mobility. The histopathology report stated that the obtained tissue was a glioblastoma, isocitrate dehydrogenase (IDH)-wildtype, WHO grade IV, O6-methylguanine-DNA methyltransferase (MGMT) un-methylated with a telomerase reverse transcriptase (TERT) mutation. The patient was referred for further oncologic treatment. 

### 3.2. Case 2

A 54-year-old female, who was operated on for a glioblastoma (GBM) in the left frontoparietal region in 2013 and afterwards underwent treatment with radiotherapy and chemotherapy, was referred by her attending oncologist. A control MRI, performed in 2021, demonstrated a lesion in the cerebellar vermis, which was enhanced after the application of contrast. As there was no growth dynamic, surgery was not suggested, and the patient was further monitored by regular MRI. On the MRI in February 2024, the lesion in the posterior fossa increased in size. It was located posteriorly to the 4th ventricle, more towards the left cerebellar hemisphere. The attending oncologist proposed that the lesion might not be GBM, therefore a stereotactic biopsy was suggested. The control CT, performed on the first postoperative day, showed a small hematoma in the biopsied location. The histopathological report of the specimen tissue stated a rosette-forming glioneuronal tumor, WHO grade I with a mutation in the fibroblast growth factor receptor 1 (FGFR1) and in the phosphatidylinositol-4,5-bisphosphate 3-kinase catalytic subunit alpha (PIK3CA). She continued with her oncological treatment and follow-ups. 

### 3.3. Case 3

A 67-year-old female presented with mild cognitive impairment and severe hyponatremia due to the syndrome of inappropriate antidiuretic hormone secretion (SIADH), which was correctly treated. Imaging diagnostic (CT and MRI of the brain) discovered a tumor in the right cerebellar hemisphere spreading towards the brainstem, and stereotactic biopsy was indicated. After surgery, the patient was transferred back to the Department of Neurosurgery where her vital signs were monitored as well as her Glasgow Coma Scale score (GCS). The patient’s condition worsened 12 h after the biopsy; she developed a severe headache and a drop in GCS was observed from 15 to 13. A control CT of the brain showed a subdural hematoma by the posterior part of the falx and around the tentorium on the left side (opposite to the site of the biopsy) with subarachnoid hemorrhage in the basal cisterns and both Sylvian fissures, spreading along the cerebellum and clivus down into the spinal canal. Following the control CT of the head, the patient went into cardiac arrest with the first rhythm being pulseless electrical activity (PEA). After 5 min of resuscitation, the return of spontaneous circulation was achieved, and she was admitted to the intensive care unit (ICU). An extraventricular draining (EVD) and intra-cranial pressure (ICP) sensor were inserted, and the first measurement was 80 mmHg; hence, a control CT of the brain was performed, which showed a large intraparenchymal hematoma around the EVD and ICP and a hematocephalus; at that point, the patient’s pupils were dilated and nonreactive. The patient went back to the theatre where a craniotomy and evacuation of the hematoma around the EVD and ICP were performed. Despite emergency surgery, the patient’s pupils remained dilated and nonreactive. The control CT of the head performed 12 h after the surgery demonstrated large hematomas in the frontal and parietal region bilaterally (not in the biopsied region) with hematocephalus. The patient required immense vasopressor support and despite the best medical treatment, ICP was slowly increasing. The patient died on the first postoperative day and an arteriovenous malformation, ruptured brain aneurysm or edema were suspected to have caused additional bleeding and not the biopsy itself. The histopathology report stated that the obtained tissue was a metastasis of melanoma. 

## 4. Technical Aspects of the Posterior Fossa Biopsy 

Each patient in our case series underwent a contrast-enhanced 3-T MRI of their brain before the decision to perform a stereotactic biopsy was made and, based on the imaging diagnostics, the most optimal biopsy trajectory was planned. On the day of the procedure, before going to the operating theatre, the Leksell Vantage headframe was positioned on the patient’s head, which was done in a room dedicated to small surgical procedures at the Department of Neurosurgery. Prior to positioning the headframe, we had to take into account the angle of the tentorium and its position in relation to the outside of the skull, so as to not interfere with our needle trajectory. When positioning the Leksell Vantage headframe for posterior fossa biopsy, we imagined a line between the inion and the patient’s external ear canal in order to be parallel with the tentorium, whereas, in cases of supratentorial biopsy, we imagine a line between the external ear canal and the cheekbone. The headframe had to be placed more towards the occipital region in order for the access window to be over the suboccipital region and local anesthetic was applied where Elekta pins were going to penetrate the skull. When putting in the pins, we had to ensure that their angle was as perpendicular to the skull as possible to provide stability to the headframe. In order to gain access to the posterior fossa, the headframe has to be positioned as low as possible, leading to pins not being perpendicular to the skull, compromising the stability of the frame; therefore, another pin can be used since the Leksell Vantage headframe has four openings on the front and back where pins can be placed to provide stability to the frame. Alternately, the existing four pins, which are used in most cases of stereotactic biopsy, can be positioned more medially. In the next step, we attempted to place the Leksell Vantage frame holder onto the headframe. However, we were unable to do so as the headframe was too low and the frame holder was pushing on the patient’s shoulders; therefore, we had to remove the headframe completely. We then joined the headframe and the frame holder and placed the full construct onto the patient’s head. By doing that, we managed to avoid pressure on the shoulders, whilst still gaining enough space to perform the biopsy. Applying the headframe with the frame holder onto the patient’s head is more challenging and requires two individuals as one cannot use the positioning band with the frame holder attached. A contrast-enhanced CT of the head with the headframe and frame holder with the Leksell Vantage CT fiducial box (Figure 1) was performed, which was merged with a preoperative planning MRI to plan the target, the entry point and the trajectory of the biopsy. Planning was performed using Medtronic’s StealthStation™ Planning Station and coordinates were calculated. When planning the trajectory, we had to take into account the frame holder and our biopsy path had to be altered as otherwise the needle would hit the frame holder. The planning and alterations had to be done manually since the software did not alert us when the frame holder was in the way of our planned trajectory (Figure 2). When setting up, we opted for the lateral right position of the stereotactic frame. A 3D model of the patient with an attached Leksell Vantage frame and the frame holder helped when setting the entry point in order to avoid the borders of the frame holder. In the operating theatre, a Leksell Vantage stereotactic arc was used to perform a stereotactic biopsy of the posterior fossa lesion. The patient, who was awake throughout the whole procedure, was in a semi-sitting position and the frame was attached to the Mayfield holder. 

Based on the initial coordinates, we discovered during surgery that the arc orientation had to be changed from lateral right to lateral left since the guide holders and the biopsy needle were hitting the frame holder. Firstly, we tried to place the patient in a full sitting position (Figure 3); however, we were still unable to get to the suboccipital region with the needle. When we changed the arc orientation from lateral right to lateral left and adjusted the stereotactic arc system to the new coordinates, we were able to access the desired region, since the guide holders and the biopsy needle bypassed the borders of the frame with the frame holder. A 5-mm skin incision was made, based on the coordinates of our trajectory, and a twist drill was placed in the desired suboccipital region in line with the planned trajectory. Eight tissue samples were obtained using a side-cutting biopsy needle (four samples at two different locations along the planned trajectory). 

Throughout the procedure, the patient was awake. After the procedure, a single suture was used to close the skin, and the patient was transferred back to the Department of Neurosurgery for further monitoring. Obtained tissue samples were subjected to a detailed histopathological and molecular neuropathological examination based on the 5th edition of the WHO Classification of Tumors of the CNS from 2021. 

In the case of patient number 1, which was a first posterior fossa lesion biopsy using the Leksell Vantage headframe, the preoperative phase comprising the positioning of the frame, a contrast-enhanced CT and planning of the biopsy trajectory on Medtronic’s StealthStation™ Planning Station took 90 min, which was considerably shorter in the second and third case at 45 and 40 min, respectively. The duration of the surgical procedure was 120 min for the first case; however, by the second case, that time was shortened to 75 min and by the third one it was down to 65 min. A correction was made during surgery only in the first case where we had to change the orientation of the frame from lateral right to lateral left as the guide holders and the biopsy needle were hitting the border of the frame. In the second and third cases, we started out with lateral left orientation of the stereotactic frame, thereby avoiding corrections during surgery and hence shortening surgery time. The results are demonstrated in Table 1. 

## 5. Leksell Vantage Headframe 

The Leksell Vantage headframe has an open-face design with two reference sides with an N-shaped design and a clear frame orientation with ANTERIOR written on top of the frame, preventing mispositioning and avoiding errors in the planning phase. The Leksell Vantage headframe has three connecting areas and is made from glass fiber-reinforced epoxy. It is suitable for a head circumference ranging from 49 to 62 cm with the width of the patient’s head being between 134 and 175 mm and the length between 167 and 215 mm. Placement of the headframe is usually done by two healthcare professionals, with one stabilizing the frame and the other fixing the pins to the head; the frame is held on by a band in the initial phase. Due to its open-face design, intubation is easier, and effective communication when the patient is awake during surgery is possible, thus making the whole surgical procedure less intimidating for patients. Due to its rigid design and a consequently limited range of positioning options, the posterior fossa lesions are challenging, since the Leksell Vantage headframe cannot be attached upside down to increase the access window, which is limited to 14 × 7 cm. As the frame is a fixed construct, the option of using longer posts to gain access to the posterior fossa is non-existent. Also, opting for a 90- or 180-degree position of the frame is not an option as it can only be positioned in one way with the only levels of freedom being the pitch, yaw and roll. However, the Leksell Vantage headframe is light, weighing only 640 g, and easy to use, with only one piece for the whole frame. 

When comparing the Leksell Vantage headframe to other commercially available systems, it is vital to recognize its advantages and shortcomings. The Cosman-Roberts-Wells (CRW) frame, which is also a rigid stereotactic frame, with graphite posts and an outer cage serving as a CT localizer, can be rotated at 90° or 180°, which can also be applied when using the Brown-Roberts-Wells frame, making posterior fossa biopsy less challenging as the positioning is easier with more options; however, the frames have several parts that have to be assembled before the biopsy takes place, thus prolonging the preoperative planning phase [23,24]. In cases of posterior fossa biopsies with the Leksell G frame, with its rectangular base ring and attached semicircular arc, the frame can be positioned upside down or longer posts can be utilized to increase the size of the biopsy window for lesions in the posterior fossa; however, the frame has many parts and the frame itself is 900 g heavier than the Leksell Vantage frame, making the procedure more demanding for the patient [25]. 

## 6. Discussion

In this article, we discuss our experiences and the learning curve with the Leksell Vantage frame for posterior fossa biopsies. The main limitation of our case series is a relatively small number of included patients, which could potentially limit the scope, depth and applicability of the current work. 

When compared to other stereotactic headframes, the Leksell Vantage frame requires more planning even before the placement of the headframe, since it is extremely rigid, can only be placed in the anteroposterior position, and allows for minimal modifications; however, the frame is light and user-friendly when it comes to assembly as it only has one component. The approximate entry point for the biopsy has to be marked on the skin, and the angle of the tentorium and the location of the transverse sinuses have to be marked out before the placement of the headframe to avoid complications. At our institution, we join the Leksell Vantage headframe with its frame holder before placing it on a patient to make sure that the placement of the headframe is feasible and that the frame does not need to be re-applied. The pins need to be as perpendicular to the skull as possible to provide us with the desired stability, making the procedure as safe and as accurate as possible. When planning the biopsy trajectory, it is important to make sure that the frame holder is not in the way of the trajectory, a detail which has to be checked manually. To make the posterior fossa biopsies feasible, we had to choose the lateral left orientation of the stereotactic arc, as in the lateral right orientation the needle was hitting the frame holder. The twist drill that was used for all posterior fossa biopsies is minimally invasive, which allows us to avoid the preparation of the muscles over the posterior fossa. In our series of stereotactic posterior fossa biopsies, patients were awake during the placement of the headframe and during the biopsy, making it easier to monitor their neurologic state, as well as avoiding complications of general anesthesia, especially in older patients with comorbidities. Despite tissue samples being small, they were representative in all cases, allowing us to provide a final diagnosis, which is vital for further personalized treatment. When performing a posterior fossa biopsy, postoperative monitoring is vital, since patients can deteriorate quickly due to relatively small bleeding or increased edema around the biopsied lesions, and quick response is imperative. 

The modified Riechert Stereotactic System (MHT Medical High Tech) that was used by our team before the Leksell Vantage headframe was less user-friendly with its numerous parts, which meant more room for human error despite a relatively good accuracy. With our previous system, we could not use the twist drill as is possible with the Leksell Vantage frame; therefore, a burr hole had to be created to gain access to the posterior fossa. The tissue samples were minute and did not allow for molecular–genetic testing, which is currently vital for an accurate diagnosis of a posterior fossa lesion according to the latest WHO CNS tumor classification. Furthermore, we also use the frameless Medtronic biopsy system based on neuronavigation; however, for that the patient has to be under general anesthesia, the procedure is more invasive with the formation of a burr hole, and it is less accurate than a stereotactic system for lesions in the posterior fossa. Based on that, our team decided to use the Leksell Vantage headframe as it allows us to perform procedures under local anesthesia, which provides us with better control over the patient’s neurological state and, with the use of a twist drill, the procedure can be minimally invasive. 

Brainstem tumors, which represent up to 10% of pediatric brain tumors, account for 1 to 2% of tumors in the adult population with approximately two-thirds located in the pons, a quarter in the medulla oblongata and up to 15% in the midbrain; however, in 80% of patients, a combination of brainstem structures is affected [26,27,28]. Lesions in the brainstem are most often glial tumors; however, metastases, lymphomas, infections or inflammation processes also have to be considered [29,30]. Up to 30% of cancer patients develop brain metastases with the most common primary site of malignancy being lung cancer (20–40%) followed by breast cancer (5–17%) and melanoma (7–11%), with posterior fossa being an important site of metastasis occurrence [31,32,33,34,35]. Multifocal lesions on MRI are often diagnosed as metastases, which is not always accurate, and tumor enhancement, seen on MRI, should also not be used to distinguish between benign and malignant lesions, since it is neither sensitive nor specific enough for low-grade neoplasms in the posterior fossa [5,36,37]. 

Due to their anatomical location, brainstem lesions manifest with cranial nerve dysfunctions in more than 80% of cases, gait disturbance in up to 61% and long-tract signs in more than 50% of cases. Signs and symptoms of increased intracranial pressure can also be found [38,39]. Posterior fossa lesions often present with a typical triad of headaches, ataxia and nausea/vomiting due to exerting mass effect and can lead to acute obstructive hydrocephalus with rapid coma and death [35]. Between 10 and 21.4% of adults with posterior fossa lesions develop hydrocephalus before surgery, and after tumor resection approximately 7% have a persistent hydrocephalus, which requires further treatment [40,41,42]. 

MRI remains the imaging of choice for posterior fossa lesions. The concordance of MRI and histopathologic evaluation of brainstem lesions is between 50 and 69%, with one study achieving an astonishing 95% concordance. However, MRI cannot distinguish between high- and low-grade tumors with sufficient accuracy: specificity is 46.6% and sensitivity is 62.5% for low-grade gliomas and 61.7% and 58.3%, respectively, for high-grade gliomas, which means that a biopsy is still required [3,30,43,44,45,46]. Great efforts are being put into developing advanced imagining technologies that would distinguish brain tumors from non-neoplastic lesions [47]. A single-voxel magnetic resonance spectroscopy (MRS), where N-Acetyl-Aspartate is typically reduced in tumors and choline is often increased, has been trialed; however, brainstem lesions—due to their small size and artefacts from surrounding tissues—remain technically difficult [48]. Additionally, ^18^F-Fluoro-Deoxy-Glucose-Positron Emission Tomography (FDG-PET) has been suggested; however, it is not widely available and it cannot distinguish areas of metabolically active brainstem tumors from highly metabolically active brain tissue with sufficient accuracy [47,48]. 

Since various diseases can occur in the posterior fossa, accurate diagnosis is crucial for further treatment and prognosis [46]. Stereotactic biopsy of posterior fossa lesions has an accuracy ranging between 87 and 100%, allowing for a deep brain target to be approached within 1 millimeter of mean positioning error with complications varying between 0 and 11% [3,5,7,11,49,50,51]. The Leksell Vantage headframe allows adjustments to the tenth of a millimeter, whereas, for instance, the ZD Inomed headframe allows for millimeter-level adjustments; however, typical stereotactic error in frame-based approaches ranges between 1 and 2 mm [21]. In cases of posterior fossa gliomas, accurate genetic profiling can be established in 89% of cases, since even when the biopsy material is non-diagnostic for histological evaluation, molecular testing can still be performed and mutations in IDH1 and TERT can be identified [5,52]. 

A stereotactic biopsy allows for a trajectory avoiding critical structures, thus reducing procedure-related complications and, therefore, being as safe and effective as supratentorial lesional biopsy [5,11]. By systematically obtaining tissue samples, the increased number of biopsies is not linked to increased complications, which can be further decreased by irrigating the biopsy site repeatedly with saline through a small plastic tube to ensure hemostasis [5,53]. According to a study conducted by Furtak et al., 95% of patients with a stereotactic posterior fossa biopsy are complication-free with 4% having minor and 1.5% having major complications including death with no significant differences in age, gender, diagnosis, location, or surgical approach [5]. Recognized risk factors for postprocedural bleeding after stereotactic biopsy are nine or more target attempts, chronic corticosteroid use, intraoperative discovery of blood within the targeting device and the presence of basal ganglial lesions, whereas deep lesion location, midline shift, lesion enhancement and histopathological nature are not significantly linked with the risk of bleeding [54]. As articles on stereotactic posterior fossa biopsy are rare, it is difficult to draw conclusions regarding the superiority of different stereotactic frame positions and approaches [21]. 

As stereotactic biopsy provides a small tissue sample, concerns have been raised that in heterogeneous lesions it may be non-representative. This issue can be overcome by collecting samples in four directions (anterior, posterior, lateral and medial) at as many points as feasible along the planned trajectory, thereby increasing the success rate of the biopsy [29,46]. In our patient series, all biopsy samples were representative and subjected to a comprehensive histological analysis and genetic mutation profiling to provide a tailored treatment plan, which was the case for patient numbers 1 and 2. 

Cephalic brainstem lesions can be accessed via the supratentorial approach, where the trajectory should avoid the ventricles to prevent the brain shifting and thus causing failure to obtain the target tissue, as well as to prevent damage to the choroid plexus and subependymal blood vessels which could lead to an intracranial hemorrhage [7,55]. In cases of caudal brainstem lesions, the supratentorial approach can be hindered by the tentorium interfering with the planned trajectory; hence, a contralateral supratentorial approach or the infratentorial transcerebellar approach should be utilized [46]. 

When it comes to the posterior fossa biopsy approach, the Leksell Vantage frame has an access window of 14 × 7 cm with a fixation attachment that further limits the posterior ring angle, thus blocking the ring range of the arc system [16]. In our case series, we overcame that by changing to the lateral left-sided approach. The rigidity of the frame’s posterior bar can interfere with the trajectory, therefore, the frame should be placed as low as possible; however, in patients with short necks and high shoulders, that is not an option [16,46]. After the first patient, we decided to place the headframe and the frame holder together, to avoid having to re-apply the frame, as we had to do with our first patient. It is crucial that the approach is tailored for each patient, taking into account the lesion location, the neuroanatomy and the idiosyncrasies of the patient’s physical attributes. Pitch, yaw and roll are the three levels of freedom for frame placement for posterior fossa biopsy, since the Leksell Vantage headframe cannot be placed upside down and can only be positioned in the anteroposterior fashion [16].

In the majority of brainstem lesions, surgical resection is not feasible except for exophytically growing lesions; therefore, contrast-enhancing as well as non-enhancing lesions in the brainstem, due to their histopathologic heterogeneity, have to be biopsied [47,56,57]. Patients that postoperatively develop obstructive hydrocephalus should be kept under surveillance since early hydrocephalus identification and management leads to longer survival, with most developing hydrocephalus in the first five-month period post-surgery [42,58]. In cases of persistent hydrocephalus, VPS or endoscopic third ventriculostomy (ETV) can be utilized in cases of limited survival prognosis with comparative efficiency [59,60,61,62]. 

Radiotherapy remains the standard treatment modality for brainstem tumors [38,47]. Stereotactic radiosurgery (SRS) represents a treatment option for posterior fossa lesions; however, it can lead to tissue swelling and mass effect on surrounding tissue, which limits its application in cases of crowded posterior fossa or impending hydrocephalus [63,64]. The role of chemotherapy remains unclear due to its poor penetrance into the CNS, especially in cases of tumor recurrence [35,48]. 

## 7. Conclusions

Posterior fossa biopsy with the Leksell Vantage frame is perhaps technically more challenging than using a different system; however, it is extremely accurate due to its rigid design. The placement of the headframe has to be planned and tailored for each patient. Despite the procedures being performed under local anesthesia, the patients need to be monitored after surgery, since minimal bleeding or an increase in perilesional edema can be life-threatening and demands instant treatment. 

Future studies should concentrate on analyzing the duration of the surgical perioperative procedures with the Leksell Vantage frame in comparison to other stereotactic headframes, as well as the role of advanced imagining technologies, such as MRI tractography, in reducing the rate of postoperative complications and new neurological deficits. 

## Figures and Tables

**Figure 1 jcm-14-00609-f001:**
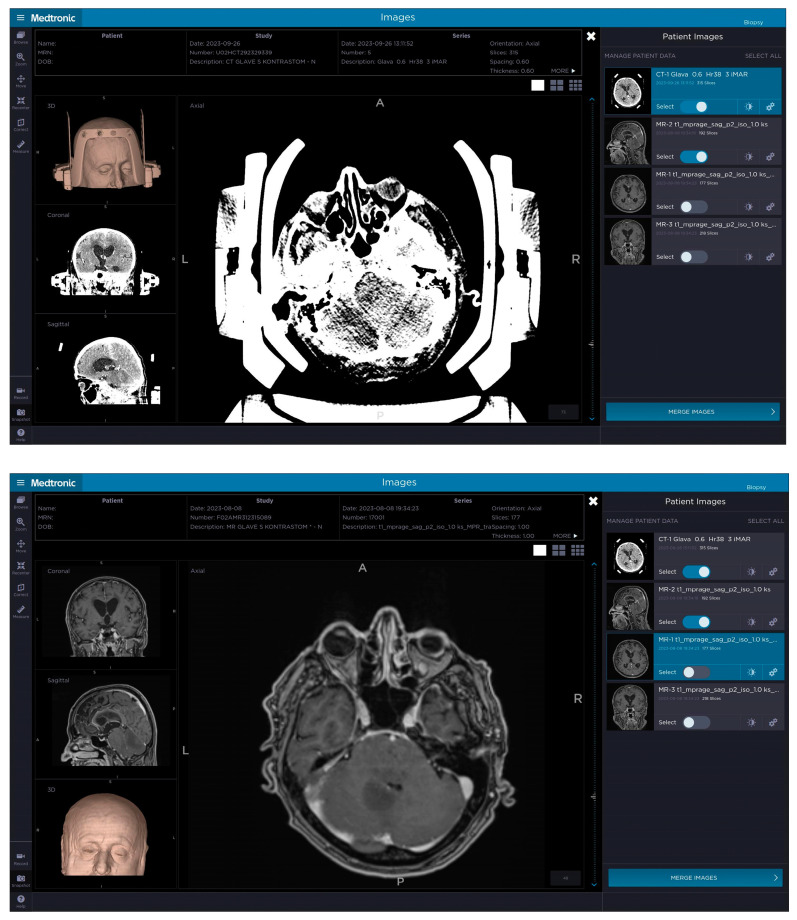
The top picture demonstrates a patient’s contrast-enhanced CT with the attached Leksell Vantage frame, frame holder and fiducial box, which was merged on Medtronic’s StealthStation™ Planning Station with the preoperative MRI (bottom picture) to plan a trajectory avoiding important anatomical structures as well as adjusting the plan so that the biopsy needle would not hit the frame and prevent the procedure. The target was set in the posterior part of the brainstem and cerebellum on the left.

**Figure 2 jcm-14-00609-f002:**
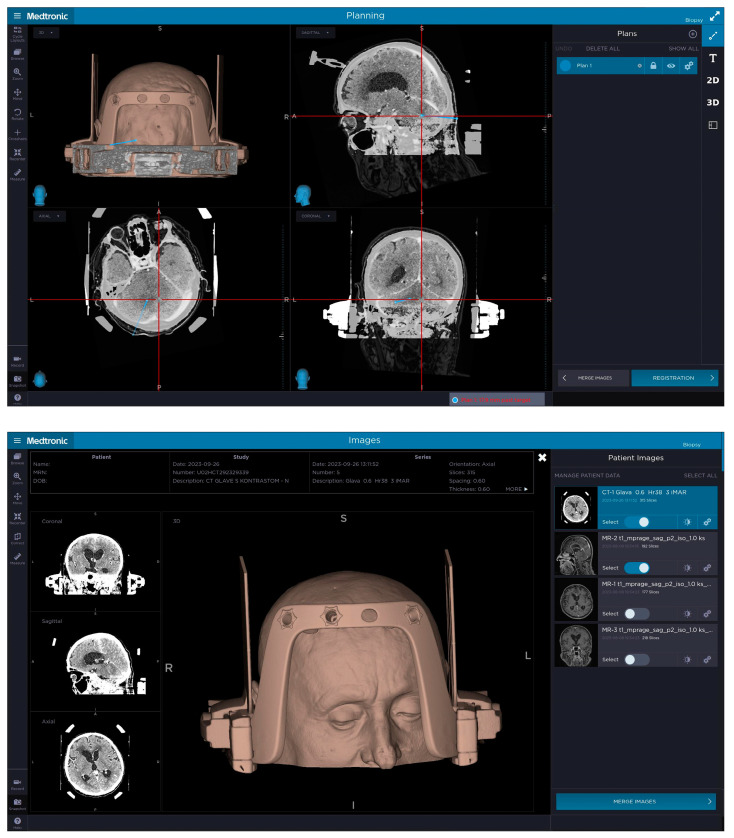
The top picture demonstrates the planned biopsy trajectory for posterior fossa biopsy for patient number 1 with the entry point and target selected, taking into account lesion location, position of patient’s shoulders and length of their neck as well as the position of the frame holder in relation to the entry point and the target. The headframe is positioned more towards the occipital region to gain access to the posterior fossa with frontal pins not being completely perpendicular to the skull. The bottom picture demonstrates a 3D model of patient number 1 with the headframe, frame holder and a CT fiducial box attached to their head, where the headframe was slightly sagittal and axially (pitch and yaw) rotated to gain access to the suboccipital region. The 3D reconstruction demonstrates the front bar of the headframe with four openings for the pins with an additional pin in place on the right side positioned more medially to gain stability.

**Figure 3 jcm-14-00609-f003:**
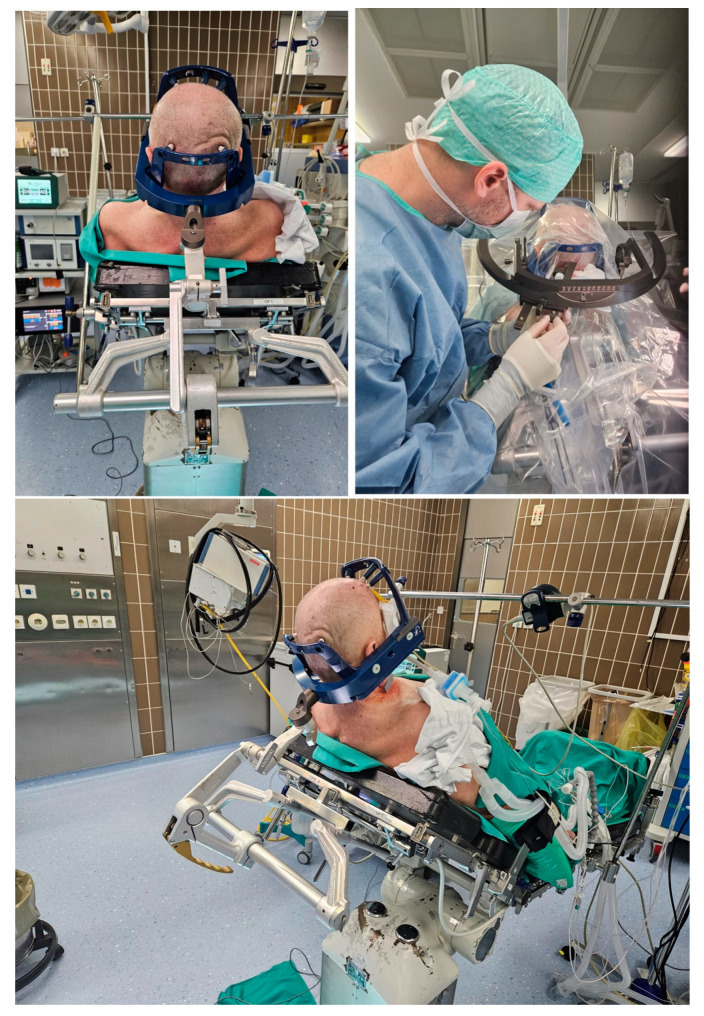
The top left picture and the lower picture demonstrate awake patient number 1 in a semi-sitting position with the rigid Leksell Vantage headframe and the frame holder attached to the patient’s head and connected to the Mayfield head holder. The top right picture demonstrates the application of the stereotactic arc system in the lateral left position (after the correction during surgery) with the biopsy needle in the guide holder passing by the borders of the frame holder.

**Table 1 jcm-14-00609-t001:** Comparing the duration of the preoperative planning and positioning of the Leksell Vantage frame, the duration of surgery and the number of corrections during operation between the three cases.

	Preoperative Planning and Positioning of the Frame	Duration of Surgical Procedure	Number of Corrections During Surgery
Patient number 1	90 min	120 min	1
Patient number 2	45 min	75 min	0
Patient number 3	40 min	65 min	0

## Data Availability

The raw data supporting the conclusions of this article will be made available by the authors on request.

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
