# Peer review of "Posterior Fossa Stereotactic Biopsy with Leksell Vantage Frame—Case Series and Review of Literature"

_jcm, 2025, doi:10.3390/jcm14020609_

Round 1
Reviewer 1 Report
Comments and Suggestions for Authors
In this paper, author s report their experience with the Leksell Vantage headframe
They wrote that stereotactic biopsy 10 using the Leksell Vantage headframe is, due to its rigid design, extremely accurate, but stiffer, making the procedure more challenging and the learning curve steeper.
However the design of this new frame was not studied, neither comparison made with its predessor.
The aim of this paper is not clear.
In conclusion section of abstract, the suggested that the rigid Leksell Vantage headframe makes access 20 to the posterior fossa tougher when compared to its predecessors, but I could not see and finding about this comparison in the study. How did authors conclude such a result.
I recomment to mention about the advantage of if there, disantantage of this frame. ,
Author Response
Comments: In this paper, authors report their experience with the Leksell Vantage headframe
They wrote that stereotactic biopsy using the Leksell Vantage headframe is, due to its rigid design, extremely accurate, but stiffer, making the procedure more challenging and the learning curve steeper.
However, the design of this new frame was not studied, and neither comparison made with its predecessor.
The aim of this paper is not clear.
In the conclusion section of the abstract, they suggested that the rigid Leksell Vantage headframe makes access 20 to the posterior fossa tougher when compared to its predecessors, but I could not see any finding about this comparison in the study. How did the authors conclude such a result?
I recommend mentioning the advantages and disadvantages of this frame.
Response: Thank you very much for your comment. We made corrections accordingly. A subheading titled “Leksell Vantage headframe” was added to the main body of the text and in that section, we described the design of the Leksell Vantage frame with a list of its advantages and disadvantages. We also provided a comparison between the Leksell Vantage frame and other stereotactic frames, such as the Cosman-Roberts-Wells frame and the Leksell G frame, providing the reader with the pros and cons of different frames as well as emphasising reasons why posterior fossa with the Leksell Vantage frame is more challenging than with other frames.
In order to make the paper’s aim clearer, we added a subtitle “Objectives” where we stated that the objective was to demonstrate the steps taken to introduce the Leksell Vantage frame into day-to-day practice and the learning curve required to master the procedure. In the subheading “Technical aspects of the posterior fossa biopsy” the reader can find information about the technical challenges we faced and how we overcame them, thus, providing valuable information for anyone planning on introducing the Leksell Vantage frame into their daily practice. In the section Discussion, we provided additional information comparing the Leksell Vantage frame with the stereotactic system with the pros and cons of each of them.
Reviewer 2 Report
Comments and Suggestions for Authors
This paper provides a detailed exploration of posterior fossa biopsies using the Leksell Vantage frame but has several areas with potential for improvement. Below are some critical areas where the paper could benefit from clarification, reorganization, and additional discussion:
• Organization: The introduction section provides valuable context but could benefit from a clearer structure. For instance, separating background, objective, and methodology into subheadings or distinct paragraphs would enhance readability.
• Case details (e.g., patient outcomes, procedural steps) are interspersed with technical discussion, which may interrupt the flow of the narrative. It could be more effective to consolidate technical observations separately.
• While figures are referenced, clearer explanations of their relevance would help readers follow procedural changes and innovations. Describing how each figure relates to procedural decisions could make the paper more coherent.
• Technical Limitations: While the paper mentions the limitations of the Leksell Vantage frame (e.g., lack of flexibility, challenging trajectory planning), it lacks a direct comparison to alternative frames. Discussing other options in more detail and specifying why the Leksell Vantage frame was chosen would give a more comprehensive context.
• The authors detail adjustments (e.g., positioning of the frame, lateral orientation) but do not fully explain why these choices are superior. Further justification or data supporting these modifications’ benefits (e.g., reduced complications, improved sample accuracy) would strengthen the argument.
• The paper mentions learning curve effects but provides little quantitative data on this. Including specific performance improvements or time savings across procedures would better highlight the adjustments’ effectiveness.
- Case 3 describes severe complications, leading to the patient’s death. The authors suggest possible non-biopsy-related causes but could explore this further, perhaps with a review of literature on posterior fossa biopsy complications. This discussion could provide balance and show a nuanced understanding of risks.
•The paper would benefit from a discussion comparing the success and accuracy rates achieved in this study with those in existing literature. For instance, noting whether the Leksell Vantage frame improved biopsy accuracy or reduced patient morbidity would add value.
- With three cases, statistical analysis is limited, yet some descriptive statistics (e.g., procedural time, number of adjustments needed) could still be reported. These would enhance the paper’s rigor and allow readers to assess the frame’s efficiency quantitatively
• The paper touches on the role of molecular diagnostics, but further discussion on how biopsy results influence personalized treatment would be beneficial. Examples of specific diagnostic protocols following biopsy results (e.g., genetic mutation profiling) could clarify this point.
• The authors could provide recommendations for teams considering the Leksell Vantage frame, such as key challenges to anticipate and ways to mitigate them.
• The paper would be strengthened by suggesting areas for future study, such as long-term outcomes for posterior fossa biopsy patients or the efficacy of newer frames with adjustable positioning features.
Author Response
Comment 1: This paper provides a detailed exploration of posterior fossa biopsies using the Leksell Vantage frame but has several areas with potential for improvement. Below are some critical areas where the paper could benefit from clarification, reorganization, and additional discussion:
- Organization: The introduction section provides valuable context but could benefit from a clearer structure. For instance, separating background, objective, and methodology into subheadings or distinct paragraphs would enhance readability.
Response 1: Thank you very much for your comment. We made corrections accordingly. We added subheadings Background, Objectives and Methodology in order to make the text clearer and more comprehensive.
Comment 2:
- Case details (e.g., patient outcomes, procedural steps) are interspersed with technical discussion, which may interrupt the flow of the narrative. It could be more effective to consolidate technical observations separately.
Response 2: Thank you very much for your comment. We made corrections accordingly. We removed technical details from the case details and moved them to the subheading “Technical aspects of the posterior fossa biopsy”.
Comment 3:
- While figures are referenced, clearer explanations of their relevance would help readers follow procedural changes and innovations. Describing how each figure relates to procedural decisions could make the paper more coherent.
Response 3: Thank you very much for your comment. We made corrections accordingly. The authors further expanded the information written underneath each picture so that the reader can get a clearer image of the steps taken in the procedure with an insight into preoperative planning and positioning of the Leksell Vantage head frame as well as the position of the patient during surgery.
Comment 4:
- Technical Limitations: While the paper mentions the limitations of the Leksell Vantage frame (e.g., lack of flexibility, challenging trajectory planning), it lacks a direct comparison to alternative frames. Discussing other options in more detail and specifying why the Leksell Vantage frame was chosen would give a more comprehensive context.
Response 4: Thank you very much for your comment. We made corrections accordingly. We added a subheading “Leksell Vantage headframe” in which we discussed the advantages and disadvantages of the Leksell Vantage headframe in cases of posterior fossa biopsy. We also provided information about other stereotactic systems used for posterior fossa biopsies with their specific pros and cons. Additionally, we discussed the systems we used before the introduction of the Leksell Vantage headframe with their advantages and disadvantages.
Comment 5:
- The authors detail adjustments (e.g., positioning of the frame, lateral orientation) but do not fully explain why these choices are superior. Further justification or data supporting these modifications’ benefits (e.g., reduced complications, and improved sample accuracy) would strengthen the argument.
Response 5: Thank you very much for your comment. We made corrections accordingly. We added additional information about the adjustments made and why they were required.
Comment 6:
- The paper mentions learning curve effects but provides little quantitative data on this. Including specific performance improvements or time savings across procedures would better highlight the adjustments’ effectiveness.
Response 6: Thank you very much for your comment. We made corrections accordingly. We added information about the duration of the preoperative planning and the duration of the surgical procedure with evidence of the shortening of both.
Comment 7: Case 3 describes severe complications, leading to the patient’s death. The authors suggest possible non-biopsy-related causes but could explore this further, perhaps with a review of the literature on posterior fossa biopsy complications. This discussion could provide balance and show a nuanced understanding of risks.
Response 7: Thank you very much for your comment. We made corrections accordingly. We added additional information on posterior fossa stereotactic biopsy complications in the Discussion section.
Comment 8:
- The paper would benefit from a discussion comparing the success and accuracy rates achieved in this study with those in existing literature. For instance, noting whether the Leksell Vantage frame improved biopsy accuracy or reduced patient morbidity would add value.
Response 8: Thank you very much for your comment. We made corrections accordingly. The Leksell Vantage headframe has comparable accuracy to other stereotactic frames, but we further discussed the advantages and disadvantages of the Leksell Vantage frame and the reasons why we decided to use it (added to the Discussion section).
Comment 9: With three cases, statistical analysis is limited, yet some descriptive statistics (e.g., procedural time, number of adjustments needed) could still be reported. These would enhance the paper’s rigour and allow readers to assess the frame’s efficiency quantitatively
Response 9: Thank you very much for your comment. We made corrections accordingly. We added information about the duration of preoperative planning and duration of the surgical procedure for each patient and the number of corrections required. It shows that even with only 3 cases, we managed to shorten preoperative times as well as time spent in the operating theatre with fewer corrections.
Comment 10:
- The authors could provide recommendations for teams considering the Leksell Vantage frame, such as key challenges to anticipate and ways to mitigate them.
Response 10: Thank you very much for your comment. We made corrections accordingly. We added information about the challenges we encountered when using the Leksell Vantage frame for the first time and how we overcame them. We also provided information on the pros and cons of the Leksell Vantage frame in comparison to other stereotactic systems as well as the system we used previously.
Comment 11:
- The paper would be strengthened by suggesting areas for future study, such as long-term outcomes for posterior fossa biopsy patients or the efficacy of newer frames with adjustable positioning features.
Response 11: Thank you very much for your comment. We made corrections accordingly. In the Conclusions section, we suggested that future studies should concentrate on the duration of the preoperative planning with the Leksell Vantage frame and compare it to other systems as well as study the role of advanced imagining technologies and their role in reducing the rate of postoperative complications.
Round 2
Reviewer 1 Report
Comments and Suggestions for Authors
The revision was well done, the paper is now acceptable
Comments on the Quality of English LanguageThe revision was well done, the paper is now acceptable